# Organohydrogel-based transparent terahertz absorber via ionic conduction loss

Wenke Xie ®[1,6], Qian Tang[1,6], Jinlong Xie[1], Yang Fei[2], Hujie Wan[1], Tao Zhao[2,3,4,5], Tianpeng Ding ®[2,3] ✉, Xu Xiao ®[2,3] ✉ & Qiye Wen ®[1,3,4] ✉

The fast-growing terahertz technologies require high-performance terahertz absorber for suppressing electromagnetic interference. Since the dissipation mechanism in terahertz band usually focuses on electronic conduction loss, almost all terahertz absorbers are constructed with electronically conducting materials being opaque, which limits their applications in scenarios requiring high visible transmittance. Here, we demonstrate a transparent terahertz absorber based on permittivity-gradient elastomer-encapsulated-organohydrogel. Our organohydrogel-based terahertz absorber exhibits a high absorbing property (average reflection loss of 49.03 dB) in 0.5–4.5 THz band with a thin thickness of 700 μm and a high average visible transmittance of 85.51%. The terahertz absorbing mechanism mainly derives from the ionic conduction loss of the polar liquid in organohydrogel. Besides, the hydrophobic and adhesive elastomer coating endows this terahertz absorber high absorbing stability and interfacial adhesivity. This work paves a viable way to designing transparent terahertz absorbers.

Terahertz (THz) wave with unique characteristics has attracted increasing research interests in recent years and promoted the rapid development of THz technologies in a wide range of applications, such as sixth generation (6 G) communication, security screening, biological sensing, astronomy detection, etc[1–3], which boosts the demands for high-performance THz absorbers to reduce the undesirable radiation and suppress the electromagnetic interference (EMI)[4]. In the past decade, various THz absorbers including carbon-based foam[5–8], MXene-based foam[9–12], metal composites[13–15], etc have been reported. Notably, almost all of them are based on electronically conducting materials, resulting from the fact that the dissipation mechanism in THz band usually focuses on electronic conduction loss[16]. However, these electronically conducting materials also possess high absorbing property in visible light band, making the THz absorbers opaque, which seriously hinders their applications in the scenarios requiring high visible transmittance, such as optical windows of aircrafts and displays of optical detecting devices[17]. Although some attempts have been made by utilizing ultrathin electronically conducting film to design metamaterial absorbers with high visible transmittance[18–20], these metamaterial absorbers suffer from narrow efficient absorption bandwidth, complicated fabrication process and high cost.

Polar liquids including water and organic solvent have been widely proved to be able to generate strong polarization loss to attenuate electromagnetic wave (EMW) in both microwave and THz bands[21–23]. Hydrogel or organohydrogel, as a class of materials constructed with three-dimensional (3D) polymer networks swollen with large amounts of water or other liquid, provides an alternative material platform for manufacturing EMW absorbers because of its ultrahigh liquid capacity[24,25]. Furthermore, compared to electrically conducting materials, hydrogels/organohydrogels exhibit high visible transparency and high mechanical stretchability[26], holding

[1]School of Electronic Science and Engineering, University of Electronic Science and Technology of China, Chengdu, China. [2]School of Physics, University of Electronic Science and Technology of China, Chengdu, China. [3]State Key Laboratory of Electronic Thin Film and Integrated Devices, University of Electronic Science and Technology of China, Chengdu, China. [4]Shenzhen Institute for Advanced Study, University of Electronic Science and Technology of China, Shenzhen, China. [5]Chengdu Research Institute, University of Electronic Science and Technology of China, Chengdu, China. [6]These authors contributed equally: Wenke Xie, Qian Tang. ✉e-mail: dingtp@uestc.edu.cn; xuxiao@uestc.edu.cn; qywen@uestc.edu.cn

great potential for providing efficient EMW absorption in optical and flexible devices. Recently, some gel-based THz absorbers have been reported[27–30]. But these THz absorbers contain large number of opaque conductive fillers, such as MXene and graphene, resulting in low visible transmittance. And the THz dissipation mechanism in these absorbers is mainly concentrated in the common electronic conduction loss. Even though some transparent EMW absorbers based on gel materials have been reported[24,31,32], these researches focused on microwave band, and the EMW dissipation mechanism of these gels is mainly attributed to polarization loss. Furthermore, it should be noted that the gel-based EMW absorbers suffer from poor absorbing stability in air and arid environment because the polar liquid that plays a key effect in attenuating EMW is susceptible to evaporation[33], which severely hinders their practical applications.

Here, we demonstrate a transparent THz absorber based on permittivity-gradient elastomer-encapsulated-organohydrogel, whose THz absorbing mechanism mainly originates from ionic conduction loss. The organohydrogel-based THz absorber is composed of a high-permittivity polyacrylamide (PAM) organohydrogel (OHG) core layer containing large amounts of water/ethylene glycol (EG) liquid, a low-permittivity polydodecyl acrylate (PDA) elastomer coating, and a PDA/OHG mixture with moderate permittivity at the PDA-OHG interface, possessing a permittivity gradient from low to high from the outside in (Fig. 1a, b). The permittivity gradient endows the PDA coated OHG (PDA@OHG) film high absorbing properties (average absorptivity of 99.81%, average reflection loss of 49.03 dB) in 0.5–4.5 THz band, with a thin thickness of 700 μm and a high average visible transmittance of 85.51% (Fig. 1d). Compared with the state-of-the-art THz absorbers, including graphene foam[6,7], MXene foam[9,12], carbon foam[8], hydrogel composite[27,28], and elastomer composite[34], our PDA@OHG film demonstrates superiority in both average reflection loss and thickness (Fig. 1c and Supplementary Table 1).

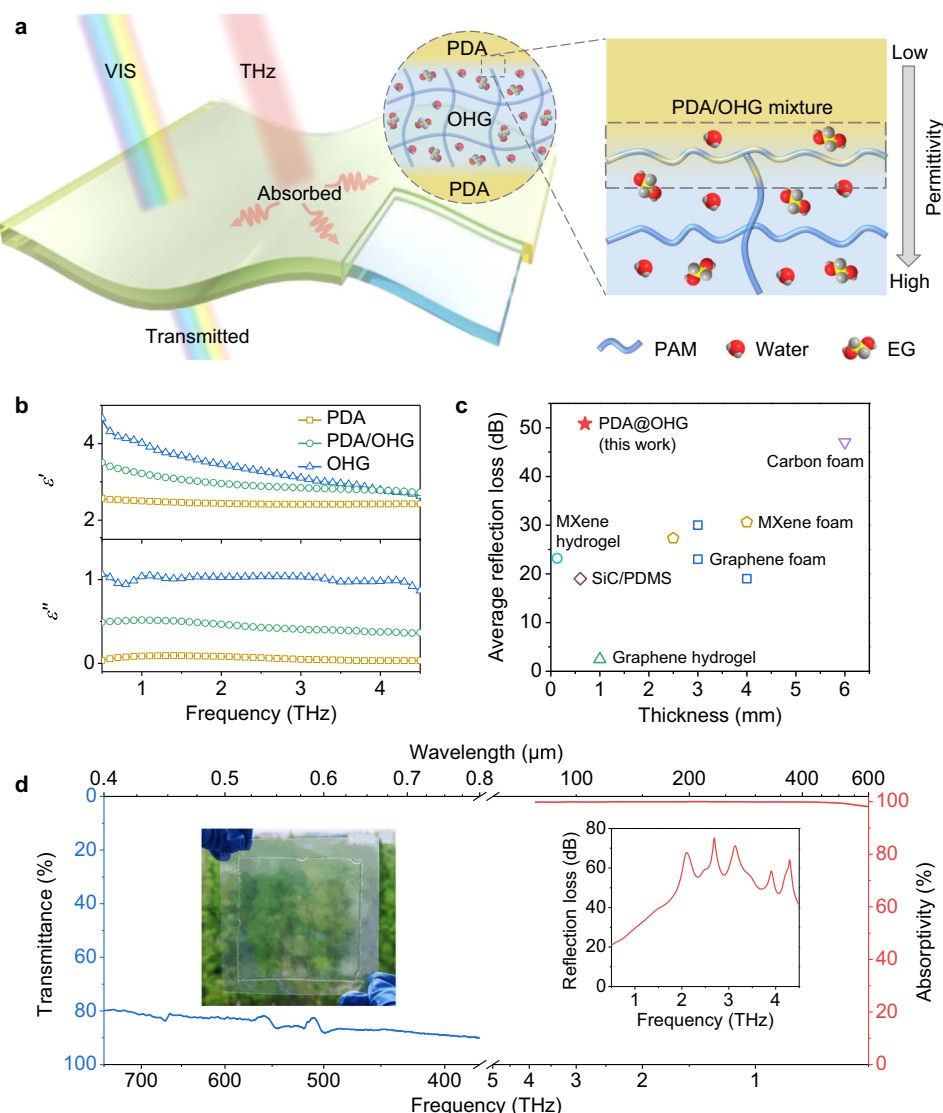

**Fig. 1 | Schematic and terahertz (THz)-visible (VIS) spectrum of polydodecyl acrylate coated organohydrogel (PDA@OHG) film. a** Schematic of the PDA@OHG film as transparent THz absorber. The gray arrow indicates the permittivity increases from the polydodecyl acrylate (PDA) layer (lower permittivity) to the organohydrogel (OHG) layer (higher permittivity). PAM: polyacrylamide, EG: ethylene glycol, red balls: oxygen, gray balls: hydrogen, yellow balls: carbon. **b** Complex permittivity of PDA, OHG, and PDA/OHG mixture in 0.5–4.5 THz band. $\varepsilon'$: real part, $\varepsilon''$: imaginary part. **c** Comparison of average reflection loss and thickness between the PDA@OHG and other THz absorbers reported in literatures. PDMS: polydimethylsiloxane. **d** Spectral visible transmittance and THz absorptivity of a 700-μm-thick PDA@OHG film. Left inset: photograph of an $80 \times 80 \times 0.7 \, mm^3$ PDA@OHG film. Right inset: reflection loss curve of the 700-μm-thick PDA@OHG film in 0.5–4.5 THz band. Source data are provided as a Source Data file.

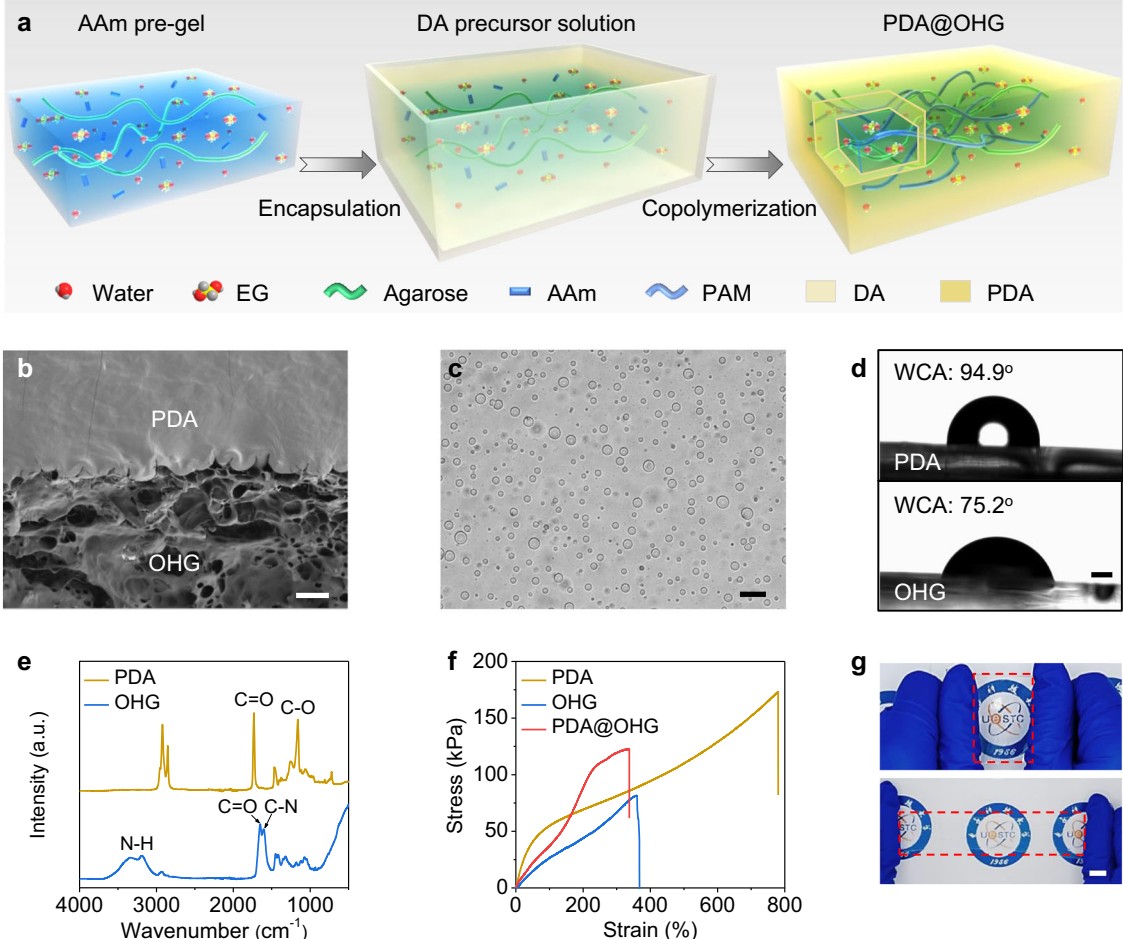

**Fig. 2 | Preparation and characterization of polydodecyl acrylate coated organohydrogel (PDA@OHG) film. a** Preparation schematic of PDA@OHG film. The cutout section of the PDA@OHG shows its core-shell structure. AAM: acrylamide, DA: dodecyl acrylate, PAM: polyacrylamide, PDA: polydodecyl acrylate, EG: ethylene glycol, red balls: oxygen, gray balls: hydrogen, yellow balls: carbon. **b** Scanning electron microscope (SEM) image of the PDA@OHG cross section. Scale bar: 50 μm. **c** Optical micrograph of the PDA-organohydrogel (OHG) interface of PDA@OHG. Scale bar: 20 μm. **d** Water contact angle (WCA) of the PDA and OHG. Scale bar: 500 μm. **e** Fourier transform infrared (FTIR) spectra of the as-prepared PDA and OHG. **f** Tensile stress-strain curves of the PDA, OHG, and PDA@OHG. **g** Photos of a PDA@OHG film in its initial and stretched state. Scale bar: 10 mm. Source data are provided as a Source Data file.

## Results and discussion

### Preparation and characterization of PDA@OHG

The PDA@OHG was prepared by a one-step copolymerization method schematically shown as Fig. 2a (see details in Methods). Firstly, an acrylamide (AAm) pre-gel consisting of AAm monomer, water/EG liquid (volume ratio = 1:1), and agarose was prepared. In which the function of agarose is to form a physical-crosslinked network to convert the AAm precursor solution into a quasi-solid state. And then the AAm pre-gel was encapsulated with dodecyl acrylate (DA) precursor solution. Finally, a PDA@OHG was obtained by the polymerization of AAm and DA monomer as well as their copolymerization at the interface (Supplementary Fig. 1). The scanning electron microscope (SEM) image of PDA@OHG cross section presents no gap at the PDA-OHG interface (Fig. 2b), indicating the successful interfacial copolymerization. And the PDA layer shows a dense morphology while the freeze-dried OHG layer shows a micrometer-scale porous morphology (Supplementary Fig. 2). Note that the original OHG has lots of nanometer-scale meshes filled with water and EG molecules[35], and the micrometer-scale pores in the SEM image are formed through a freeze-drying process. In addition, there is a PDA/OHG mixture region at the PDA-OHG interface, which is confirmed by the optical micrograph of PDA-OHG interface showing numerous microspheres with diameters of 1–15 μm (Fig. 2c) caused by the immiscibility of the hydrophobic PDA and the hydrophilic OHG (Fig. 2d).

The chemical composition of the as-prepared PDA@OHG was analyzed by energy dispersive spectroscopy (EDS) and Fourier transform infrared (FTIR) spectroscopy. The EDS data shows carbon, oxygen, and nitrogen elements in the OHG region, while only carbon and oxygen elements in the PDA region (Supplementary Fig. 3). In the FTIR spectrum of PDA region, the peaks located at 1732 cm$^{-1}$ and 1160 cm$^{-1}$ can be attributed to the stretching of C = O and C-O, which are characteristic peaks of the acrylate[36]. And the FTIR spectrum of the OHG region shows absorption signals at 1602 cm$^{-1}$, 1648 cm$^{-1}$, and 3337 cm$^{-1}$, corresponding to C-N shearing, C = O stretching, and N-H stretching, respectively, which are characteristic peaks of the amide[37] (Fig. 2e). The above results indicate the successful synthesis of the PDA@OHG via the copolymerization method. And the conversion rate of the AAm and DA are both as high as >98% by using this synthesis method (Supplementary Fig. 4). Furthermore, our PDA@OHG exhibits high tensile property with an elongation of 337.42% and a breakage strength of 122.35 kPa (Fig. 2f, g), which benefits from the double network structure of the OHG and the high mechanical property of the PDA. Note that double network structure is a famous strategy to improve the mechanical properties of gels[38]. Here, the OHG is composed of a rigid physically-crosslinked agarose network and a soft chemically-crosslinked PAM network. And the agarose network could dissipate lots of stress energy during deformation process to increase the fracture energy of OHG.

## THz absorbing and visible transmitting properties of PDA@OHG

The ideal EMW absorber should have good impedance matching and strong EMW attenuation capacity simultaneously, which can be achieved in the PDA@OHG with permittivity gradient structure. The impedance matching value ($Z_c$) of the low-permittivity PDA is about 0.67 in 0.5–4.5 THz band (Supplementary Fig. 5), a little lower than 1, meaning good impedance matching with air media. The good impedance matching of PDA is beneficial to allow the entrance of THz wave into the PDA@OHG. And the PDA/OHG mixture with moderate impedance matching value can further increase the transmission of THz wave from PDA layer to OHG layer. The high-permittivity OHG has high attenuation coefficient ranging from 51.34 cm$^{-1}$ to 523.76 cm$^{-1}$ in 0.5–4.5 THz band (Supplementary Fig. 6), implying that the OHG could efficiently dissipate the penetrated THz wave. As a result, our PDA@OHG exhibits high absorbing property in frequency range of 0.5–4.5 THz. After thickness optimization of the OHG layer and the PDA coating (Supplementary Figs. 7, 8), the average absorptivity of the PDA@OHG with an optimized thickness of 700 μm reaches 99.81% in 0.5–4.5 THz range, and its reflection loss (RL) exceeds 20 dB over the whole measured frequency range, with a high average RL of 49.03 dB (red line and right inset in Fig. 1d). Note that the volume ratio of water: EG in the OHG is 1:1. As the EG ratio is negatively correlated with the THz permittivity of the organohydrogel (Supplementary Fig. 9), the hydrogel (HG) containing pure water (EG ratio of 0%) has higher THz permittivity than the OHG (EG ratio of 50%) and PDA. Thus, the PDA coated HG (PDA@HG) also has a similar permittivity gradient structure to the PDA@OHG, which endows the PDA@HG high absorbing properties (average absorptivity of 99.87%, average reflection loss of 44.62 dB) in 0.5–4.5 THz band (Supplementary Fig. 10).

However, the PDA@OHG and the PDA@HG have completely different transmittance in visible light band (wavelength range of 0.4–0.8 μm). The PDA@OHG exhibits high average transmittance of 85.51% in visible light band, while the visible transmittance of PDA@HG is below 20% over the whole band (blue line in Fig. 1d, Supplementary Fig. 11), which is caused by the inherent optical properties of the PDA, OHG and HG in visible light band. In fact, the visible transmittance of a composite is affected by the extinction coefficient ($k$) and refractive index ($n$) of its components. Specifically, a low extinction coefficient means a high transmittance, and the close refractive index between components is beneficial to suppress the scattering of visible light at the interface, making the composite have high visible transmittance[39]. In terms of the extinction coefficient, the PDA, OHG, and HG all have low extinction coefficient of less than 0.01 in visible light band, thus their average visible light transmittance is both more than 85% (Supplementary Fig. 12). In terms of the refractive index, by comparing the refractive index of the organohydrogel at different EG ratio, we find that the refractive index of the OHG (EG ratio of 50%) is closest to that of the PDA, while the refractive index of the HG (EG ratio of 0%) is much lower than that of the PDA (Supplementary Fig. 13). As a result, the PDA@OHG exhibits high visible light transmittance while that of the PDA@HG is low. In addition, the THz absorbing and visible transmitting properties of the PDA@OHG are also affected by the microsphere structure at the PDA-OHG interface. Compared with the uniform structure, the microsphere structure can slightly increase the THz absorption while slightly decreasing the visible light transmittance (Supplementary Fig. 14).

## THz absorbing mechanism of PDA@OHG

In order to reveal the THz absorbing mechanism of PDA@OHG, the THz absorbing properties of the PDA, OHG, and corresponding freeze-dried PAM aerogel with the same thickness of 500 μm were firstly compared. As shown in Fig. 3a, the OHG exhibits high absorptivity ranging from 87.81% to 94.78% in the whole 0.5–4.5 THz range, while the PDA shows a poor absorbing property with the lowest absorbtivity of only 20.72%. And the simulated attenuation ability represented by

power loss density (PLD) of OHG is much higher than that of PDA (Supplementary Fig. 15). The experimental and simulated results both proves that the OHG layer plays a dominant role in attetuating the incident THz wave in PDA@OHG. The freeze-dried PAM aerogel has a poor absorbing property with the lowest absorbtivity of only 17.13% in 0.5–2 THz range, which means the water/EG liquid in the OHG plays a key role in dissipating the THz wave. Note that the high absorptivity of the PAM aerogel in 2–4.5 THz range is caused by the synergistic effect of the 3D porous structure with the PAM material, which is verified by the slightly lower absorptivity of the dense PAM film than that of the porous PAM aerogel in 2–4.5 THz range (Supplementary Fig. 16).

To further ascertain the THz absorbing mechanism of the OHG, the conductivity ($\sigma$), relaxation time ($\tau$), conduction loss ($\varepsilon''_c$), and polarization loss ($\varepsilon''_p$) of the OHG and corresponding PAM aerogel were calculated by their permittivity spectra based on Debye theory[40] (Supplementary Fig. 17, see details in Supplementary Note 3). Computational results demonstrate that for both OHG and PAM aerogel, conduction loss plays a dominant role while polarization loss contributes little (Fig. 3b and Supplementary Fig. 18). The small proportion of polarization loss is mainly caused by their high relaxation time of 72.92 ps and 51.96 ps (Supplementary Fig. 19), meaning the resonance frequencies ($f_{res} = \frac{1}{2\pi\tau}$) of the polarization are located at 2.18 GHz and 3.06 GHz respectively, much far from THz band. The large proportion of conduction loss derives from the high conductivity of several to hundreds of Siemens per meter in 0.5–4.5 THz band, which is several orders of magnitude higher than the conductivity in 10$^{-2}$-10$^6$ Hz band (Fig. 3c and Supplementary Fig. 20). It worth noting that the conductivity curve in Fig. 3c is a combination of two frequency bands, which are obtained by two different measurement methods. The conductivity in 10$^{-2}$-10$^6$ Hz band is measured by electrochemical impedance spectroscopy (EIS) (see details in Supplementary Fig. 21), while the conductivity in 0.5–4.5 THz band is extracted from the THz time-domain spectra (TDS) based on Debye theory. It should be mentioned that the PAM aerogel is almost completely insulated so that its impedance and conductivity can't be measured by EIS. The conductivity of OHG and PAM aerogel exhibits a strong frequency dependence over the measured frequency range, whereas ordinary electronic conductors such as mentals and carbon usually possess flat conduvtivity in wide frequency range from Hz band to THz band[4,41]. In addition, the conductivity of OHG and PAM aerogel (here taking the conductivity at 0.8 THz as example) increases with the increasement of temperature (Fig. 3d), while electronic conductors usually have negative temperature coefficient. And the relationship between conductivity and temperature fits well with Arrehenius equation $\sigma = \frac{\sigma_0}{T}\exp(-\frac{E}{kT})$), which is usually adopted to describe the temperature dependence of ionic conductivity[42]. Therefore, it can be inferred that the conduction loss in the OHG is mainly ionic conduction loss rather than common electronic conduction loss.

Based on the above results, we propose an ionic-conduction-loss-dominated THz absorbing mechanism for the PDA@OHG absorber. Firstly, the THz wave enters into the PDA layer without significant reflection due to the good impedance matching. Then the penetrated THz wave propagates in the OHG layer, in which a large part of THz wave is dissipated by the osciliations of ions (including anions and cations) which originate from the dissociation of water/EG and the foreign ions in PAM polymer[43], and a small part of THz wave is dissipated by the polarization relaxation of water and EG molecules (Fig. 3e).

## Absorbing stability and adhesion property of PDA@OHG

For gel-based EMW absorbers containing large amounts of polar liquid, absorbing stability is a critical challenge in practical applications. Because the polar liquid that plays a dominant role in attenuating EMW is susceptible to evaporation in air and arid environment,

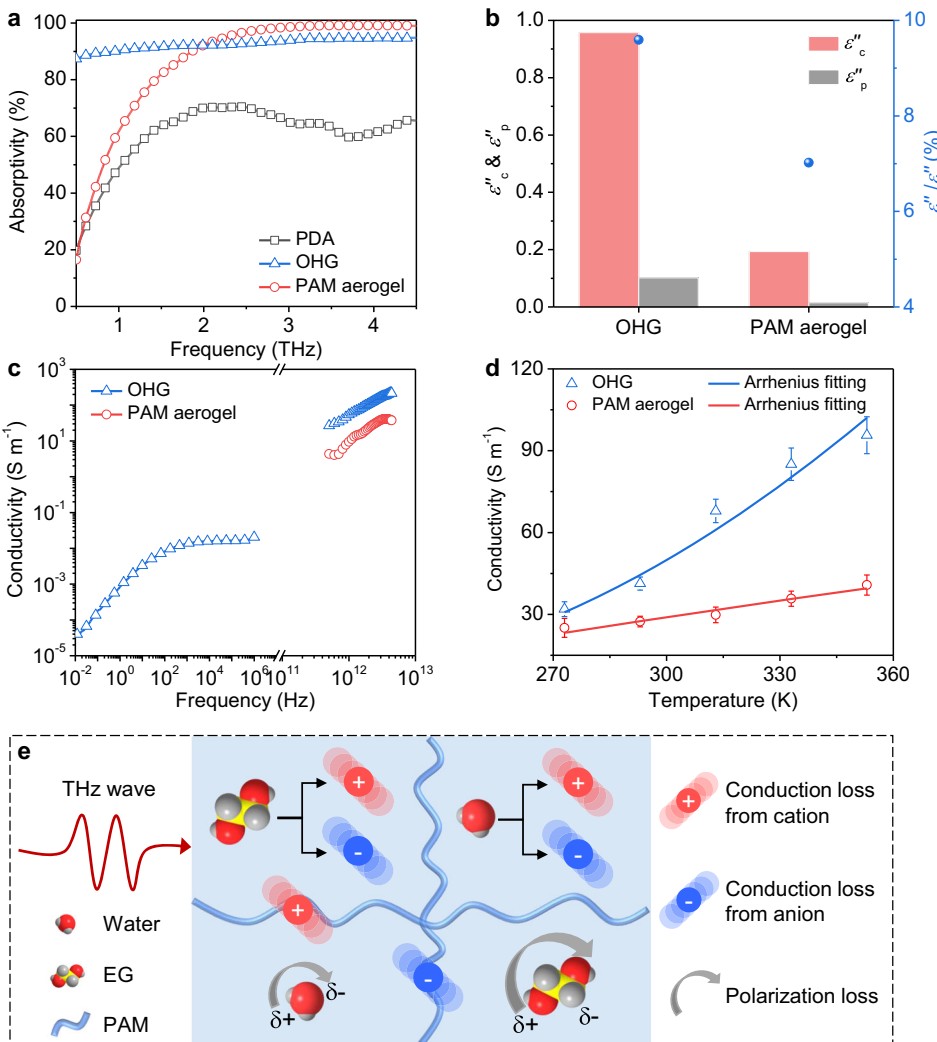

**Fig. 3 | Terahertz (THz) absorbing mechanism of polydodecyl acrylate coated organohydrogel (PDA@OHG). a** Absorptivity curves of the 500-μm-thick polydodecyl acrylate (PDA), organohydrogel (OHG), and polyacrylamide (PAM) aerogel in 0.5–4.5 THz band. **b** Conduction loss ($\varepsilon''_c$), polarization loss ($\varepsilon''_p$), and the ratio of polarization loss ($\varepsilon''_p/\varepsilon''$) of the PDA, OHG, and PAM aerogel. **c** Conductivity spectra of the OHG and PAM aerogel in frequency range of $10^{-2}$ Hz to 4.5 THz. Note that there is a break in the frequency range of $10^6$ Hz to 0.5 THz. And it should be mentioned that the PAM aerogel is insulated in $10^{-2}$–$10^6$ Hz so that its conductivity in $10^{-2}$-$10^6$ Hz can't be measured. **d** Conductivities of the OHG and PAM aerogel at 0.8 THz as a function of temperature. Each error bar represents the standard deviation of 5 measurements. **e** Schematic illustration of the THz absorbing mechanism of the OHG. EG: ethylene glycol, red balls: oxygen, gray balls: hydrogen, yellow balls: carbon. There are various ions in the OHG, which mainly originate from the dissociation of water and EG molecules, and a small part is the foreign ions from PAM polymer. When THz wave acts on the OHG, these ions are oscillated to generate strong conduction loss, while the water and EG molecules also generate polarization loss to attenuate the THz wave. Source data are provided as a Source Data file.

which can decrease the EMW absorbing properties. For instance, a $50 \times 50 \times 0.5$ mm³ HG with pure water as dispersion rapidly lost almost all of water (weight ratio of 69.10%) within 3 days at air conditions of 25 °C and 30 RH%, and the OHG containing high-boiling-point EG still lost 16.80% of its weight in 4 days (Supplementary Fig. 22). In consequence, the average reflection loss of HG decreased from 9.61 dB to 5.95 dB after 3 days (Supplementary Fig. 23), and the average reflection loss of OHG decreased from 10.94 dB to 6.37 dB after 4 days (Supplementary Fig. 24). In contrast, the PDA@OHG exhibits much improved anti-drying property and THz absorbing stability at the same condition, thanks to the dense and hydrophobic PDA coating. As shown in Fig. 4a, the PDA@OHG only lost 4.60% of its weight within 15 days caused by the little loss of water, making a slight reduction in average reflection loss. Nevertheless, the PDA@OHG still has a high average reflection loss of ~40 dB after 15 days, and demonstrates efficient THz absorption (reflection loss > 10 dB) over the entire measured frequency range (Supplementary Fig. 25).

Apart from the absorbing stability, another important function for EMW absorber, is the adhesivity to substrates. To quantify the adhesivity of PDA@OHG, we carried out a standard 90-degree peeling test with a peeling rate of 50 mm min⁻¹, the adhesion energy is equal to the steady-state peeling force per width (see details in Supplementary Fig. 26). The measured adhesion energy on steel, copper, polyethylene terephthalate (PET), polymethyl methacrylate (PMMA), silicon, and glass is 189.24 J m⁻², 229 J m⁻², 136 J m⁻², 163 J m⁻², 153 J m⁻², and 154 J m⁻², respectively (Fig. 4b), almost comparable to that of commercial tape[41]. Insets in Fig. 4b show the peeling process of the PDA@OHG on the corresponding substrates, in which the PDA@OHG near interface front became highly deformed and developed into the typical yield region, further indicating the strong adhesivity to substrates. The strong adhesivity can be attributed to two reasons: one is the inherent high adhesivity of PDA coating that is a commonly used acrylate adhesive, and the other is the high energy dissipation ability of the OHG caused by the double-network structure[44].

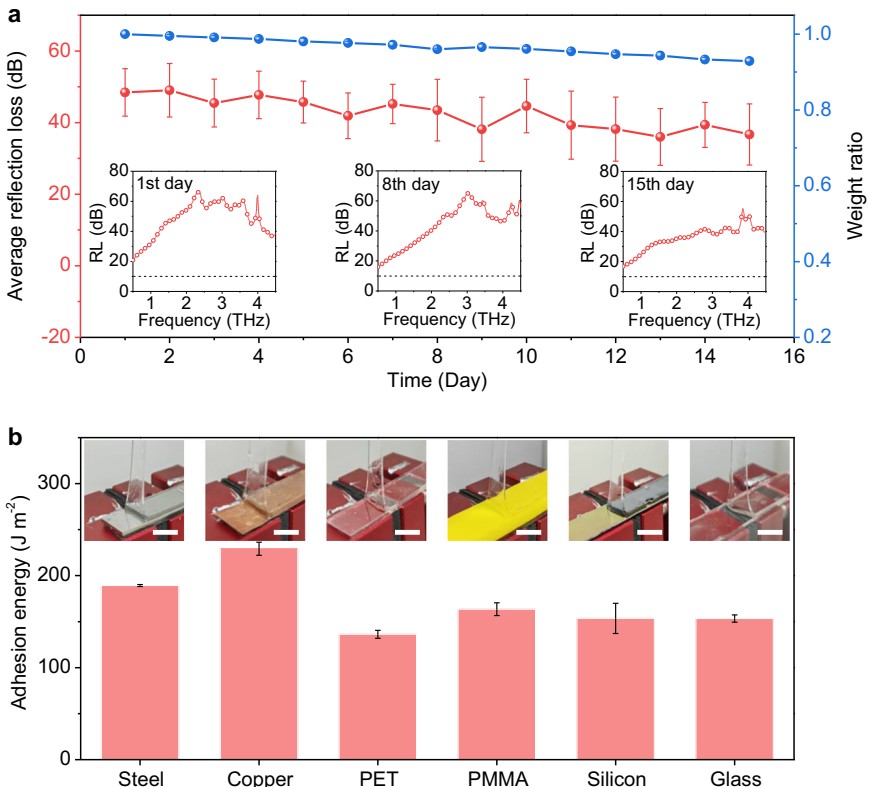

**Fig. 4 | Absorbing stability and adhesion property of polydodecyl acrylate coated organohydrogel (PDA@OHG). a** Changes in average reflection loss (RL) and weight ratio of the PDA@OHG with storage time at air conditions of 25 °C and 30 RH%. Each error bar represents the standard deviation of 5 measurements. Insets: RL curves of the PDA@OHG on the 1st, 8th, and 15th day. Each black dashed line in the inset represents the commercial standard of effective terahertz absorption. **b** Adhesion energy of the PDA@OHG on diverse substrates. PET: polyethylene terephthalate, PMMA: polymethyl methacrylate. Each error bar represents the standard deviation of 5 measurements. Insets: photographs of the 90-degree peeling process for PDA@OHG on the corresponding substrates. Scale bars: 10 mm. Source data are provided as a Source Data file.

In summary, this work reports a transparent THz absorber based on permittivity-gradient elastomer-encapsulated-organohydrogel (PDA@OHG), which is prepared by a one-step copolymerization method. The PDA@OHG shows a high average reflection loss of 49.03 dB in 0.5–4.5 THz band with a thin thickness of 700 μm and a high average visible transmittance of 85.51%. The THz absorbing mechanism mainly originates from the ionic conduction loss of water/ EG liquid. In addition, the PDA@OHG has high absorbing stability and adhesion property benefiting from the hydrophobic and adhesive PDA coating. Our work paves ways to designing transparent absorbers that may advance terahertz applications and beyond.

## Methods
### Chemicals and materials
Acrylamide (AAm), N,N′-methylenebis(acrylamide) (MBA), 2-hydroxy-4′-(2-hydroxyethoxy)-2-methylpropiophenone, dodecyl acrylate (DA), ethylene glycol dimethacrylate (EGDMA), 2,2-diethoxyacetophenone, ethylene glycol (EG), and agarose were purchased from Aladdin Chemical Co. in China. All reagents were used as received. Milli-Q (18.3 MΩ) water was used in all experiments.

### Preparation of organohydrogel (OHG) and polyacrylamide (PAM) aerogel
The OHG was prepared by a typical radical polymerization method. An AAm precursor solution consisting of water/EG binary solvent (volume ratio = 1:1), 2 mol L$^{-1}$ AAm as monomer, 0.1 wt% MBA as crosslinker, 0.1 wt% 2-hydroxy-4′-(2-hydroxyethoxy)-2-methylpropiophenone as initiator, and 2 wt % agarose as physically-crosslinked network was firstly obtained by heating at 95 °C for 30 min. Then the AAm precursor

solution was poured into a glass mold and cooled at 8 °C for 30 min to obtain a physically-crosslinked AAm pre-gel. Finally, the double-network OHG was obtained by making the AAm pre-gel under ultraviolet (UV) irradiation (wavelength of 365 nm, power density of 4 mW cm$^{-2}$) for 1 h. The PAM aerogel was obtained by freeze-drying the above OHG.

### Preparation of polydodecyl acrylate coated organohydrogel (PDA@OHG)
The PDA@OHG was prepared by a one-step copolymerization method. An AAm pre-gel which is same with OHG, and a DA precursor solution consisting of DA solvent as monomer, 1 wt% ethylene glycol dimethacrylate as crosslinker, and 0.1 wt% 2,2-diethoxyacetophenone as initiator was firstly prepared. The AAm pre-gel was immersed in the DA precursor solution in an encapsulated glass mold. And a PDA@OHG was simply obtained by irradiating the glass mold containing DA precursor solution and AAm pre-gel with 365 nm UV light (power density of 4 mW cm$^{-2}$) for 1 h.

### Materials characterizations
The morphology was observed by field emission scanning electron microscopy (SEM) (FEI, Nova Nano SEM 450). The elemental distribution was analyzed by using the energy dispersive spectrometer (EDS) mapping in SEM. Before this test, the sample was brittle failure at liquid nitrogen and then freeze-dried. The chemical bond was characterized by Fourier-transform infrared (FTIR) spectroscopy (Bruker, TENSOR27). The mechanical property was determined by tensile tests using a universal mechanical test machine (REGER, RGM-4005T). The adhesion property was measured using the standard 90-degree

peeling test (ASTM, D2861) with a mechanical testing machine and a 90-degree peeling fixture. The transmittance in visible light was measured by an UV-VIS (visible) spectrometer (Shimadzu, UV-3600). The complex refractive index was obtained from the ellipsometry (J. A. Woollam, IR-VASE).

## Measurement of terahertz absorbing property

The terahertz (THz) absorbing property was obtained by using a THz time domain spectrometer (TDS) system with both reflection and transmission modes. The effective frequency range was 0.5–4.5 THz. The reflection loss (RL) values of the samples were calculated according to the following equation:

$$RL = -20 \log(E_s/E_r) \qquad (1)$$

Where $E_s$ and $E_r$ refers to the amplitude of the reflection signal for the sample and the aluminum plate, respectively.

## Calculation of polarization loss and conduction loss

The polarization loss ($\varepsilon_p''$) and conduction loss ($\varepsilon_c''$) were calculated based on Debye theory[40]:

$$\varepsilon' = \varepsilon_\infty + \frac{\varepsilon_s - \varepsilon_\infty}{1 + \omega^2\tau^2} \qquad (2)$$

$$\varepsilon'' = \frac{\varepsilon_s - \varepsilon_\infty}{1 + \omega^2\tau^2}\omega\tau + \frac{\sigma}{\omega\varepsilon_0} \qquad (3)$$

$$\varepsilon_p'' = \frac{\varepsilon_s - \varepsilon_\infty}{1 + \omega^2\tau^2}\omega\tau \qquad (4)$$

$$\varepsilon_c'' = \frac{\sigma}{\omega\varepsilon_0} \qquad (5)$$

where $\omega$ is the angular frequency, $\varepsilon_s$ is the stationary dielectricconstant, $\varepsilon_\infty$ is the optical dielectric constant, $\sigma$ is the electrical conductivity, $\tau$ is the relaxation time. Then the non-linear squares fitting method was used to fit the relaxation time, conductivity, polarization loss and conductive loss (see Supplementary Note 3 for detail).

## Data availability

Source data are provided with this paper. All other relevant data supporting the findings of this study are available from the corresponding authors on request. Source data are provided with this paper.

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

## Acknowledgements

This work is supported by the National Key Research and Development Program of China (grant numbers 2023YFB3811303 (X.X.), 2023YFB3811305 (Q.W.), 2020YFA0714001 (T.Z.)), the National Natural Science Foundation of China (grant numbers 62235004 (Q.W.), 61831012 (Q.W.), 62311530115 (Q.W.), 52202084 (T.D.), and 62375044 (T.Z.)), the China Postdoctoral Science Foundation (grant number 2022M710605 (W.X.)), the Outstanding Scholarship Foundation of UESTC (grant number A1098531023601345 (T.D.)), the Sichuan Science and Technology Support Program (grant numbers 2021JDTD0026 (Q.W., X.X.), 2023JDGD0012 (Q.W.)), the Natural Science Foundation of Sichuan Province (grant numbers 2023NSFSC0437 (X.X.), 2023NSFSC0959 (T.D.), and 2022NSFSC0514 (T.Z.)), the Fundamental Research Funds for the Central Universities (grant number ZYGX2020J003 (T.Z.)), the Special Grants for Postdoctoral Research Projects in Sichuan Province (grant number TB2023031 (T.Z.)).

## Author contributions

Q.W., X.X., and T.D. supervised the work. W.X. and Q.T. carried out the experimental measurements. J.X. performed the terahertz time domain measurements of different temperatures. W.X., Q.T., T.D., X.X., and Q.W. analyzed the data. Y.F., H.W., and T.Z. assisted in the analysis. W.X., T.D., X.X., and Q.W. wrote the manuscript with input from all co-authors.

## Competing interests

The authors declare no competing interests.
