## [Peer Review File · Nature Communications]

Reviewers' Comments:

Reviewer #1:

Remarks to the Author:

This manuscript provided detail information about the organohydrogel-based THz absorber, it could be published after revision.

1. the title is not easy to understand.
2. there are related papers published, what is the different of this research and others?
3. what is the reason for the formation of porous of the OHG?
4. what is the conversion of AAm and DA? unreacted monomer will affect the properties of the film
5. the Tg of PDA is not very high, is it possible that PDA will gradually transfer into the porous of the OHG?
6. the boiling point of EG is high enough to make sure it will not easy to evaporate, but water will evaporated slowly, what is effect of water lose for the organohydrogel-based THz absorber?
7. what is the thickness of the whole film? because the penetration of UV light is limited, it will affect the polymerization.
8. the references should keep the same format.
9. the English need improvement.

Reviewer #2:

Remarks to the Author:

In this work, the authors demonstrate a THz absorber based on permittivity-gradient elastomer-encapsulated-organohydrogel. This visibly transparent THz absorber differs from conventional electronically conducting material based THz absorber, which is quite novel and interesting. Authors have done comprehensive experiments to explain the THz absorption mechanism clearly, and ascribe it to ionic conduction loss. Overall, this work is innovative, and would be interested by the communities of both the soft materials and EMI shielding. Thus, I would like to recommend the publication of this manuscript in Nature Communications. However, there are still some minor corrections need to be addressed, as below.

1. As the HG with pure water as dispersion liquid has higher permittivity than the OHG containing water/EG liquid in THz band, while the refractive index of the HG is lower than that of the OHG in visible light. Does the ratio of water/EG in the OHG have effect on the permittivity and refractive index? Please provide some experimental data and discussion on this question.
2. How did the authors obtain the dielectric and optical properties such as complex permittivity, impedance matching value, attenuation coefficient, and refractive index? Authors should provide the detailed measurement and calculation methods of these parameters.
3. The authors have provided simulations results in the Supporting Information, however, the parameters are not given. Please provide the simulation details.
4. Whether the microspheres shown in Fig. 2c have any effect on visible light transmittance and terahertz absorption?
5. The conductivity curve in Fig. 3c is incomplete, why? How about the conductivity of PDA@OHG?

Reviewer #3:

Remarks to the Author:

This paper reported an interesting organohydrogel-based film for THz absorption. The organohydrogel-based film exhibits a high absorptivity in 0.5-4.5 THz band and a high transparency in visible light band. Authors have done comprehensive work to show the key advantages and characteristics of the THz absorbing film. They have also deeply analyzed the THz absorbing mechanism of the film, and proposed a novel mechanism based on ionic conduction loss. This manuscript can be accepted provided the following questions are answered and some

revisions are made as suggested below:

- 1) As shown in Fig. 2a, except from AAm polymer, agarose polymer is also contained in the OHG layer. What role does the agarose play? Please give a more detailed discussion in the text.
- 2) The authors mentioned that the OHG has double network structure, resulting in a high mechanical property. What is the double network structure? And why is the double network structure beneficial to improve the mechanical properties? The explanation of the double network should be added.
- 3) Fig. 3a shows that the PAM aerogel has a high absorptivity of nearly 100% in 2-4.5 THz range. Is this caused by the three-dimensional porous structure of the aerogel? Many reported electromagnetic wave absorbers have the similar porous structures, which can increase the transmission path and losses of the penetrated electromagnetic waves. I suggest to add some discussions about it.

Response to referees

We would like to express our sincere thanks to reviewers for the critical and careful review of the entire manuscript. Their valuable comments and constructive suggestions are helpful for us to reword some of the statements in the text to make it precise, clear and easy to read, and have illuminated us to further polish our research with highlighted academic value. Listed below are our point-by-point responses to reviewers' questions/comments.

Reviewer #1:

General Comments: This manuscript provided detail information about the organohydrogel-based THz absorber, it could be published after revision.

Response: Thanks for the reviewer's positive comments. We have revised the manuscript according to reviewer's suggestion and added some related references.

Comment 1: The title is not easy to understand.

Response: We thank the reviewer for this kind reminder. To make the title easier to understand, we have changed the title to "*Organohydrogel-based transparent terahertz absorber via ionic conduction loss*".

Comment 2: There are related papers published, what is the different of this research and others?

Response: We appreciate the reviewer's close attention to the field. Indeed, there are some papers about gel-based THz absorbers. For instance, Zhu et al. reported a $Ti_3C_2T_x$ MXene hydrogel toward absorption-dominated electromagnetic-interference (EMI) shielding in 0.2-2.0 THz band (*ACS Nano* **2021**, 15, 1465-1474), Zhang et al reported a reduced graphene oxide/ γ -GY (RGO/GY) heterostructures aerogel for enhanced electromagnetic wave absorbing properties in both gigahertz and terahertz band (*Nano Res.* **2023**, 16, 88-100). However, **these gel-based THz absorbers contain large number of opaque conductive fillers**, such as MXene and graphene, which results in low visible transmittance. **And the THz dissipation mechanism in these absorbers mainly focused on the common electronic conduction loss from conductive fillers.** Even though transparent electromagnetic wave (EMW) absorbers based on gel materials have been reported, (for example, Zhao et al. reported a series of controllable

microwave absorbers based on transparent hydro/organo/ionogels-*Adv. Mater.* **2022**, 34, 2205376, Song et al. reported an optically transparent PVA hydrogel for optically manipulatable microwave stealth structures-*Adv. Sci.* **2020**, 7, 1902162), **these researches focused on microwave band, and the EMW dissipation mechanism of these gels is mainly concentrated in polarization loss.** In our work, we **firstly demonstrate a transparent and conductive-filler-free THz absorber based on permittivity-gradient elastomer-encapsulated-organohydrogel, and we propose a new THz dissipation mechanism based on ionic conduction loss.** Overall, our work differs from the reported works in both materials and mechanism.

We have added more discussion about the difference between our work and other works in **Line 55-62, Page 2-3** in the **Revised Manuscript** as follow:

“Recently, some gel-based THz absorbers have been reported²⁷⁻³⁰. But these THz absorbers contain large number of opaque conductive fillers, such as MXene and graphene, resulting in low visible transmittance. And the THz dissipation mechanism in these absorbers is mainly concentrated in the common electronic conduction loss. Even though some transparent EMW absorbers based on gel materials have been reported^{24, 31, 32}, these researches focused on microwave band, and the EMW dissipation mechanism of these gels is mainly attributed to polarization loss.”

Comment 3: What is the reason for the formation of porous of the OHG?

Response: We thank the reviewer for the valuable question. The OHG is a three-dimensional crosslinked PAM network with large amounts of water/EG liquid as the dispersed medium. The mesh size of the polymer network is actually only ~10 nm (*Soft Matter* **2013**, 9, 5483; *Eng. Fract. Mech.* **2018**, 187, 74), and the meshes are filled with water and EG molecules (**Fig. R1**). The micrometer-scale pores in the SEM image are formed through the freeze-drying process, because the original OHG containing volatile liquid can't be directly characterized by SEM. In another word, the SEM image shows the morphology of freeze-dried PAM aerogel rather than the original morphology of OHG. Before SEM characterization, the sample is firstly frozen in liquid nitrogen and then freeze-dried in a freeze dryer. During the freezing process, lots of water/EG crystals form and grow. And then in the freeze-drying process, the crystals sublimate to form several micrometer-scale pores.

To make the manuscript much clearer, a discussion about the structure of OHG have been added in **Line 101-103, Page 5** in the **Revised Manuscript** as follow:

“Note that the original OHG has lots of nanometer-scale meshes filled with water and EG molecules³⁵, and the micrometer-scale pores in the SEM image are formed through

a freeze-drying process.”

Fig. R1. Schematic of the structure of OHG before and after freeze drying.

Comment 4: What is the conversion of AAm and DA? Unreacted monomer will affect the properties of the film.

Response: To eliminate the reviewer’s concerns on the effects of the unreacted monomer on the properties of film, we have conducted the experiments to measure the polymers’ conversion rate. The conversion rate is obtained by measuring the weight ratio of the polymerized sample to the original monomer. The OHG sample and PDA sample were firstly prepared through the polymerization method. Note that the OHG sample was polymerized under a 100- μm -thick PDA layer to mimic the polymerization conditions of the PDA@OHG film. Then the as-prepared samples were dried in a vacuum drying oven at 60 °C for 24 h to obtain the mass of m_1 (this drying process can be omitted for the PDA sample). After that the OHG sample and PDA sample were respectively immersed in pure water and ethyl acetate for 24 h to remove the unreacted monomers, and then they were dried again in the vacuum drying oven at 60 °C for 24 h to obtain the mass of m_2 . The conversion rate equals m_2/m_1 . As shown in **Fig. R2**, the measured conversion rates of AAm and DA are as high as 98.59% \pm 0.09% and 98.17% \pm 0.12%, respectively (each error represents the deviation from three data points), indicating that there are very few unreacted monomers in the PDA@OHG film.

The result has been added in **Supplementary Fig. 4** in the **Revised Supplementary**

Information, and the corresponding discussion has been added in **Line 118-119, Page 5-6** in the **Revised Manuscript** as follow:

“And the conversion rate of the AAm and DA are both as high as >98% by using this synthesis method (Supplementary Fig. 4).”

Fig. R2. Conversion rate of AAm and DA.

Comment 5: The Tg of PDA is not very high, is it possible that PDA will gradually transfer into the porous of the OHG?

Response: We thank the reviewer for this question. Even though the Tg of PDA is low, it is difficult for PDA to transfer into the OHG. OHG is hydrophilic with filling by water/EG liquid, however, the PDA elastomer is hydrophobic, making them difficult to mix together. In addition, the pore size in the OHG network is only in nanometer scale, which can also prevent the polymerized PDA to transfer into the OHG. Nevertheless, before the polymerization, the DA liquid can partially mix with the AAM pre-gel at the interface to form a PDA/OHG mixture region (**Fig. R3**).

Fig. R3. Schematic diagram of the PDA-OHG interface before and after polymerization.

Comment 6: The boiling point of EG is high enough to make sure it will not easy to evaporate, but water will evaporate slowly, what is effect of water lose for the organohydrogel-based THz absorber?

Response: We thank the reviewer for this meaningful comment. Since water plays an important role in absorbing THz wave, the evaporation of water will decrease the THz absorption for the gel-based absorbers. As shown in **Fig. R4**, a $50 \times 50 \times 0.5 \text{ mm}^3$ hydrogel with pure water as dispersion rapidly lost almost all of water (weight ratio of 69.10%) within 3 days at air conditions of $25 \text{ }^\circ\text{C}$ and 30 RH%, making the average reflection loss decrease from 9.61 dB to 5.95 dB. However, our organohydrogel-based THz absorber has a good water retention capacity, thanks to the dense and hydrophobic PDA coating. Although the addition of EG can increase the water retention capacity to a certain extent, the OHG with water/EG liquid while without PDA coating lost 16.80% of its weight in 4 days, leading to a reduction of average reflection loss from 10.94 dB to 6.37 dB. In contrast, the PDA@OHG exhibits much improved anti-drying property and THz absorbing stability at the same condition. the PDA@OHG only lost 4.60% of its weight within 15 days caused by the little loss of water, making a slight reduction in the average reflection loss. As a result, the PDA@OHG still has a high average reflection loss of $\sim 40 \text{ dB}$ after 15 days. Moreover, the anti-drying property of the PDA@OHG can be further improved by increasing the ratio of EG.

The discussion about the effect of the water evaporation on the THz absorption have been added in **Line 256-258, Page 11** in the **Revised Manuscript** as follow:

“As shown in Fig. 4a, the PDA@OHG only lost 4.60% of its weight within 15 days caused by the little loss of water, making a slight reduction in average reflection loss.”

Fig. R4. a, b Change in weight ratio (a) and average reflection loss (b) of HG, OHG and PDA@OHG with storage time under environmental conditions ($25 \text{ }^\circ\text{C}$, humidity 40%).

Comment 7: What is the thickness of the whole film? Because the penetration of UV light is limited, it will affect the polymerization.

Response: We agree that the penetration of UV light is important to the polymerization. **Fig. R5a** shows the cross-sectional optical micrograph of the PDA@OHG film, from which we can see that the thickness of the whole film is about 700 μm . The wavelength of the UV light we used is 0.365 μm , and the transmittance of the PDA@OHG film at the wavelength of 0.365 μm is as high as 85.82% (**Fig. R5b**). The result indicates the UV light could penetrate through the PDA@OHG film and induce the polymerization sufficiently, which is also verified by the high conversion rate of AAm and DA (Fig. R2).

Fig. R5. a Cross-sectional optical micrograph of the PDA@OHG film. **b** Transmittance curve of the PDA@OHG film in UV (wavelength of 0.34-0.40 μm) band.

Comment 8: The references should keep the same format.

Response: We thank the reviewer for pointing this out. We have adapted the format of all references to the requirements of the journal.

Comment 9: The English need improvement.

Response: We thank the reviewer for reading our paper carefully. Regarding the language of the whole manuscript, we have checked the paper thoroughly and asked an English major friend to make improvements in the revised version.

Reviewer #2:

General Comments: In this work, the authors demonstrate a THz absorber based on permittivity-gradient elastomer-encapsulated-organohydrogel. This visibly transparent THz absorber differs from conventional electronically conducting material based THz absorber, which is quite novel and interesting. Authors have done comprehensive experiments to explain the THz absorption mechanism clearly, and ascribe it to ionic conduction loss. Overall, this work is innovative, and would be interested by the

communities of both the soft materials and EMI shielding. Thus, I would like to recommend the publication of this manuscript in Nature Communications. However, there are still some minor corrections need to be addressed, as below.

Response: We appreciate that the reviewer affirms the novelty of our work. We have revised the manuscript according to reviewer's suggestion and added several references.

Comment 1: As the HG with pure water as dispersion liquid has higher permittivity than the OHG containing water/EG liquid in THz band, while the refractive index of the HG is lower than that of the OHG in visible light. Does the ratio of water/EG in the OHG have effect on the permittivity and refractive index? Please provide some experimental data and discussion on this question.

Response: We highly appreciate the reviewer's valuable question and constructive suggestion. We have measured the THz permittivity and visible refractive index of the organohydrogel with different EG ratio (0%, 25%, 50%, 75% and 100%). For the permittivity in THz band, as the EG ratio increases from 25% to 50%, the average real part of permittivity decreases from 5.16 to 2.50, and the average imaginary part of permittivity decreases from 2.21 to 0.60 (**Fig. R6**). For the refractive index in visible light band, its average value increases from 1.38 to 1.48 with the increasement of the EG ratio from 0% to 75%, but decreases to 1.39 at the EG ratio of 100% (**Fig. R7**). Note that the average visible refractive index of the PDA is 1.46. To make the PDA@OHG film have high THz absorption and high visible transmittance simultaneously, the organohydrogel layer should have high THz permittivity and similar visible refractive index with the PDA layer. Therefore, 50% is an optimal EG ratio for the organohydrogel layer.

The results have been added in **Supplementary Fig. 9** and **Supplementary Fig. 13** in the **Revised Supplementary Information**. And the related discussions have also been added in **Line 151-154, Page 7** and **Line 170-174, Page 8** in the **Revised Manuscript** as follow:

“Note that the volume ratio of water : EG in the OHG is 1:1. As the EG ratio is negatively correlated with the THz permittivity of the organohydrogel (Supplementary Fig. 9), the hydrogel (HG) containing pure water (EG ratio of 0%) has higher THz permittivity than the OHG (EG ratio of 50%) and PDA.”

“In terms of the refractive index, by comparing the refractive index of the organohydrogel at different EG ratio, we find that the refractive index of the OHG (EG ratio of 50%) is closest to that of the PDA, while that of the HG (EG ratio of 0%) is

much lower than that of the PDA (Supplementary Fig. 13).”

Fig. R6. **a, b** Real part (a) and imaginary part (b) of complex permittivity of the organohydrogel with different EG ratio in 0.5-4.5 THz band. **c** Average permittivity of the organohydrogel as a function of EG ratio in 0.5-4.5 THz band.

Fig. R7. **a** Refractive index of the organohydrogel with different EG ratio in visible light band. **b** Average refractive index of the organohydrogel as a function of EG ratio in visible light band. The dash line represents the average refractive index of the PDA.

Comment 2: How did the authors obtain the dielectric and optical properties such as complex permittivity, impedance matching value, attenuation coefficient, and refractive

index? Authors should provide the detailed measurement and calculation methods of these parameters.

Response: The THz dielectric/optical parameters are extracted from the transmitted THz signal of the sample ($\tilde{E}_{\text{sam}}(\omega)$) and the reference ($\tilde{E}_{\text{ref}}(\omega)$), which are Fourier transformed from the time-domain measurements. The optical parameters including refractive index $n(\omega)$ and extinction coefficient $k(\omega)$ are firstly deduced from the complex transfer function $\tilde{H}(\omega)$ via:

$$\tilde{H}(\omega) = \frac{\tilde{E}_{\text{sam}}(\omega)}{\tilde{E}_{\text{ref}}(\omega)} = \frac{4n(\omega)}{[1+n(\omega)]^2} \cdot \exp\left\{-k(\omega) \frac{\omega d}{c}\right\} \cdot \exp\left\{-i[n(\omega) - 1] \frac{\omega d}{c}\right\} \quad (1)$$

Where d is the thickness of sample, ω is the angular frequency of THz wave, c is the vacuum speed of light. Based on the refractive index and extinction coefficient, the complex permittivity $\tilde{\epsilon}(\omega)$ and the attenuation coefficient $\alpha(\omega)$ are calculated from the following equations:

$$\tilde{\epsilon}(\omega) = \epsilon'(\omega) + i\epsilon''(\omega) = (n(\omega) + ik(\omega))^2 \quad (2)$$

$$\alpha(\omega) = \frac{2\omega k(\omega)}{c} \quad (3)$$

The impedance matching value $Z_c(\omega)$ is obtained by:

$$Z_c(\omega) = \sqrt{\mu/\tilde{\epsilon}(\omega)} \quad (4)$$

Where μ is the permeability, which equals 1 here. And the visible refractive index and extinction coefficient are measured by ellipsometry (J. A. Woollam IR-VASE).

To make our work much clearer, the method of THz dielectric parameter extraction has been added in **Supplementary Note 1** in the **Revised Supplementary Information**.

Comment 3: The authors have provided simulations results in the Supporting Information, however, the parameters are not given. Please provide the simulation details.

Response: The commercial software COMSOL Multiphysics 5.4 (COMSOL, Stockholm, Sweden) was used to simulate the THz absorption of the materials in alternating electromagnetic fields. A radio frequency (RF) physics field was applied in the finite element simulation process. A 0.5×1 cm rectangular sandwiched by two 0.1×1 cm rectangular with measured electromagnetic parameters and ionic conductivities were designed to represent the PDA@OHG. Perfectly matched conditions were imposed on the air domain to eliminate interference from the reflected

waves. Two ports (port 1 and port 2) were established in the air domain to calculate the S-parameters. Port 1 was activated with 1 W m^{-1} input power in electric field mode. The COMSOL simulation was conducted in the steady analysis mode. We have added the description of finite element simulation in **Supplementary Note 2** in the **Revised Supplementary Information**.

Comment 4: Whether the microspheres shown in Fig. 2c have any effect on visible light transmittance and terahertz absorption?

Response: We are grateful for this valuable question. To investigate the effects of the microspheres on visible transmittance and terahertz absorption, two PDA@OHG samples with and without microspheres are prepared by two different preparation methods. Sample #1 is obtained by one-step copolymerization method, which has microspheres at the PDA-OHG interface (**Fig. R8a, b**). And sample #2 is obtained by assembling the polymerized PDA layer on the polymerized OHG layer, which has no microsphere at the PDA-OHG interface (**Fig. R8c, d**). In terms of the visible transmittance, sample #2 without microspheres has a higher visible transmittance than sample #1 with microspheres (**Fig. R8e**). This is because the microspheres create lots of PDA-OHG interfaces, which can increase the scattering of visible light. In terms of the THz absorption, sample #1 with microspheres has 1% higher absorptivity than sample #2 without microspheres (**Fig. R8f**), which is caused by the lower THz reflectance of the sample #1 with microspheres (**Fig. R8g**).

The results have been added in **Supplementary Fig. 14** in the **Revised Supplementary Information**, and the corresponding discussion has been added in **Line 175-179, Page 8** in the **Revised Manuscript** as follow:

“In addition, the THz absorbing and visible transmitting properties of the PDA@OHG are also affected by the microsphere structure at the PDA-OHG interface. Compared with the uniform structure, the microsphere structure could slightly increase the THz absorption while slightly decreasing the visible light transmittance (Supplementary Fig. 14).”

Fig. R8. **a, c** Preparation schematic of the sample #1 with microspheres (a) and the sample #2 without microspheres (c). **b, d** Optical micrograph of the PDA-OHG interface of the sample #1 (b) and sample #2 (d). **e** Transmittance of the sample #1 and sample #2 in visible light band. **f, g** Absorptivity (f) and reflectance (g) of the sample #1 and sample #2 in 0.5-4.5 THz band.

Comment 5: The conductivity curve in Fig. 3c is incomplete, why? How about the conductivity of PDA@OHG?

Response: The conductivity curve in Fig. 3c is a combination of the conductivity of two frequency bands, which are obtained by two different measurement methods. The conductivity in 0.01-10⁶ Hz band is measured by electrochemical impedance spectroscopy (EIS), which is obtained by $\sigma = L/(A \times R)$. Where L is thickness, A is area, R is the resistance (real part of the impedance). While the conductivity in 0.5-4.5 THz band is extracted from the THz time-domain spectra based on Debye theory, and the effective frequency range of the THz time-domain spectrometer system is 0.5-4.5 THz. The conductivities in 10⁶ Hz-4.5 THz band are not measured, so that the

conductivity curve is incomplete.

Since the PDA coating of the PDA@OHG is insulated in the low frequency range (0.01-10⁶ Hz), the conductivity of the PDA@OHG in 0.01-10⁶ Hz range is too low to be measured. While in the high frequency range (0.5-4.5 THz), the conductivity of all samples increases significantly. For the PDA@OHG, its conductivity varies from 13.01 S m⁻¹ to 93.54 S m⁻¹, which is between the conductivity of PDA and OHG (**Fig. R9**).

Fig. R9. Conductivity curves of the PDA@OHG, PDA, and OHG in 0.5-4.5 THz range.

To make our paper clearer, a discussion about the conductivity extraction has been added in **Line 217-222, Page 10** in the **Revised Manuscript** as follow:

“It worth noting that the conductivity curve in Fig. 3c is a combination of two frequency bands, which are obtained by two different measurement methods. The conductivity in 0.01-10⁶ Hz band is measured by electrochemical impedance spectroscopy (EIS) (see details in Supplementary Fig. 21), while the conductivity in 0.5-4.5 THz band is extracted from the THz time-domain spectra based on Debye theory.”

Reviewer #3:

General Comment: This paper reported an interesting organohydrogel-based film for THz absorption. The organohydrogel-based film exhibits a high absorptivity in 0.5-4.5 THz band and a high transparency in visible light band. Authors have done comprehensive work to show the key advantages and characteristics of the THz absorbing film. They have also deeply analyzed the THz absorbing mechanism of the film, and proposed a novel mechanism based on ionic conduction loss. This manuscript can be accepted provided the following questions are answered and some revisions are made as suggested below:

Response: We appreciate that the reviewer affirms the novelty of our work. We have revised the manuscript according to reviewer's suggestion and added several references.

Comment 1: As shown in Fig. 2a, except from AAm polymer, agarose polymer is also contained in the OHG layer. What role does the agarose play? Please give a more detailed discussion in the text.

Response: Thanks for the reviewer's valuable question and constructive suggestion. The addition of agarose is an important step in the one-step copolymerization method. Because agarose can transform the AAm precursor solution into a quasi-solid state with a certain shape by forming physical-crosslinked networks, and then chemical-crosslinked PAM networks are formed in the agarose matrix by polymerization process. Furthermore, the agarose can also enhance the mechanical properties of the OHG, which benefits from the double network structure. We have added the discussion about the function of agarose in **Line 93-94, Page 5** in the **Revised Manuscript** as follow:
“In which the function of agarose is to form a physical-crosslinked network to convert the AAm precursor solution into a quasi-solid state.”

Comment 2: The authors mentioned that the OHG has double network structure, resulting in a high mechanical property. What is the double network structure? And why is the double network structure beneficial to improve the mechanical properties? The explanation of the double network should be added.

Response: Double network structure is a famous strategy to improving the mechanical properties of gels. Here, the OHG is composed of a rigid physically-crosslinked agarose network and a soft chemically-crosslinked PAM network. During the large deformation process, the agarose network is gradually sacrificed to dissipate a lot of stress energy, resulting in an increasement in fracture energy. We have added a detailed discussion about the double network structure in **Line 122-126, Page 6** in the **Revised Manuscript** as follow:

“Note that double network structure is a famous strategy to improve the mechanical properties of gels³⁸. Here, the OHG is composed of a rigid physically-crosslinked agarose network and a soft chemically-crosslinked PAM network. And the agarose network could dissipate lots of stress energy during deformation process to increase the fracture energy of OHG.”

Comment 3: Fig. 3a shows that the PAM aerogel has a high absorptivity of nearly 100% in 2-4.5 THz range. Is this caused by the three-dimensional porous structure of the

aerogel? Many reported electromagnetic wave absorbers have the similar porous structures, which can increase the transmission path and losses of the penetrated electromagnetic waves. I suggest to add some discussions about it.

Response: To explore the effect of the structure of the PAM aerogel on its THz absorption, the THz absorption of a freeze-dried PAM aerogel with porous structure and a vacuum-dried PAM film with dense structure at the same weight were compared. As shown in **Fig. R10**, the absorptivity of PAM film in 2-4.5 THz range varies from 82.69% to 96.37%, a little smaller than that of PAM aerogel (92.35%-99.11%). The result indicates that the 3D porous structure of the PAM aerogel could increase the THz absorption, but the high absorptivity of the PAM aerogel in 2-4.5 THz range is mainly caused by the PAM material. We have added the result in **Supplementary Fig. 16** in the **Revised Supplementary Information**, and the related discussion has also been added in **Line 192-195, Page 8-9** in the **Revised Manuscript** as follow:

“Note that the high absorptivity of the PAM aerogel in 2-4.5 THz range is caused by the synergistic effect of the 3D porous structure with the PAM material, which is verified by the slightly lower absorptivity of the dense PAM film than that of the porous PAM aerogel in 2-4.5 THz range (Supplementary Fig. 16).”

Fig. R10. Absorptivity curves of the porous PAM aerogel and dense PAM film with same weight in 0.5-4.5 THz band.

Reviewers' Comments:

Reviewer #1:

Remarks to the Author:

The authors have answered all the questions. The manuscript could be published by nature communication.

Reviewer #2:

Remarks to the Author:

The authors have revised their manuscript carefully according to referee's comments. I would like to recommend the acceptance of this paper for publication.

Reviewer #3:

Remarks to the Author:

it has been revised well and i think it can be accepted

Response to referees

We would like to express our sincere thanks to reviewers for the careful review of the entire manuscript. Listed below are our point-by-point responses to reviewers' comments.

Reviewer #1:

General Comments: The authors have answered all the questions. The manuscript could be published by nature communication.

Response: Thanks for the reviewer's positive comments.

Reviewer #2:

General Comments: The authors have revised their manuscript carefully according to referee's comments. I would like to recommend the acceptance of this paper for publication.

Response: We appreciate the reviewer for the positive comments.

Reviewer #3:

General Comment: it has been revised well and i think it can be

Response: Thanks for the reviewer's positive comments.